# Controlled Release of 18-β-Glycyrrhetinic Acid from Core-Shell Nanoparticles: Effects on Cytotoxicity and Intracellular Concentration in HepG2 Cell Line

**DOI:** 10.3390/ma14143893

**Published:** 2021-07-12

**Authors:** Giuseppina Nocca, Giuseppe D’Avenio, Adriana Amalfitano, Laura Chronopoulou, Alvaro Mordente, Cleofe Palocci, Mauro Grigioni

**Affiliations:** 1Dipartimento di Scienze biotecnologiche di base, Cliniche Intensivologiche e Perioperatorie, Università Cattolica del Sacro Cuore, 00168 Rome, Italy; adriana.amalfitano@unicatt.it (A.A.); alvaro.mordente@unicatt.it (A.M.); 2Fondazione Policlinico Universitario A. Gemelli IRCCS, 00168 Rome, Italy; 3National Center for Innovative Technologies in Public Health, Istituto Superiore di Sanità, 00161 Rome, Italy; giuseppe.davenio@iss.it (G.D.); mauro.grigioni@iss.it (M.G.); 4Department of Chemistry, University La Sapienza, 00185 Rome, Italy; cleofe.palocci@uniroma1.it; 5CIABC-Centro di Ricerca per le Scienze Applicate alla Protezione dell’Ambiente e dei Beni Culturali, University La Sapienza, 00185 Rome, Italy

**Keywords:** nanoparticles, drug delivery, ECIS, cytotoxicity

## Abstract

18β-glycyrrhetinic acid (GA) is a pentacyclic triterpene with promising hepatoprotective and anti-Hepatocellular carcinoma effects. GA low water solubility however reduces its biodistribution and bioavailability, limiting its applications in biomedicine. In this work we used core-shell NPs made of PolyD-L-lactide-co-glycolide (PLGA) coated with chitosan (CS), prepared through an osmosis-based methodology, to efficiently entrap GA. NPs morphology was investigated with SEM and TEM and their GA payload was evaluated with a spectrophotometric method. GA-loaded NPs were administered to HepG2 cells and their efficiency in reducing cell viability was compared with that induced by the free drug in in vitro tests. Cell viability was evaluated by the MTT assay, as well as with Electric Cells-Substrate Impedance Sensing (ECIS), that provided a real-time continuous monitoring. It was possible to correlate the toxic effect of the different forms of GA with the bioavailability of the drug, evidencing the importance of real-time tests for studying the effects of bioactive substances on cell cultures.

## 1. Introduction

Hepatocellular carcinoma (HCC) is a major cause of cancer-related death worldwide. The progress in both surgical and nonsurgical treatments has induced significant benefits in terms of survival [1]. However, we are still a long way from having an effective drug without side shortcomings, such as poor solubility and low specificity and biodistribution. The need for an HCC therapy based on the selectivity of the molecules towards specific target receptors is evident.

18β-glycyrrhetinic acid (GA) is a pentacyclic triterpene derived from the hydrolysis of glycyrrhizin (GZ), a pentacyclic triterpene glycoside extracted from *Glycyrrhiza glabra*. GA is associated to a plethora of biological activities, among which hepatoprotective and anti-HCC effects. This latter ability of GA is due to multiple mechanisms, such as cell cycle arrest [2], induction of autophagy and apoptosis [3,4] and so on. The specificity of GA versus hepatocytes is very high because it is based on the interaction with specific receptors on the liver [5]: GA-R (GA-Receptor) or GZ-R (GZ- Receptor). Previous studies have shown that the receptors of the first type are more numerous than the second ones on hepatocytes, suggesting that GA may be much more effective than GZ [6]. Moreover, GA induced cell cycle arrest in the G1-phase and apoptosis in HepG2 (human hepatocarcinoma) cells. Such effect is probably due both to a decrease of Bcl-2 and Bcl-XL (anti-apoptotic proteins) and to the activation of caspases-8 and -9 [2].

Unfortunately, GA has a poor water solubility and this characteristic can reduce its biodistribution and bioavailability. A way to solve these issues is to convey GA directly into the cancer cells using specific carriers. In fact, drug delivery through nanovectors is gaining more and more interest due to the possibility of releasing the active compound in a targeted and controlled way [7].

Moreover, drug delivery allows maintaining the therapeutic concentration, for an extended period of time, within the cells. Numerous studies, in fact, report significant differences in the effect of the same active ingredient when it is delivered by using a nanovector, compared to when it is not [8]. These differences may relate to the actual intracellular concentration of the drugs or the rapid onset of the toxic event.

Polymeric nanoparticles (NPs) have attracted interest as delivery vehicles for different compounds, with the potential to overcome issues such as poor drug solubility, cell permeability and unspecific cytotoxicity [9].

Core-shell NPs of PolyD-L-lactide-co-glycolide (PLGA) coated with chitosan (CS) have an excellent safety profile, good biocompatibility, low levels of immunogenicity and toxicity and tunable in vivo biodegradation rate [10,11]. Moreover, PLGA and CS have been approved by the US Food and Drug Administration (FDA) and European Medicines Agency (EMA) as effective carriers for drug delivery in humans [12,13]. Due to the differences observed between vehiculated and non-vehiculated drug effects, it is very interesting to monitor cell proliferation in real-time to precisely identify the onset of the toxic event. The classic systems for evaluating in vitro toxicity—Trypan blue, MTT (3-(4,5-dimethylthiazol-2-yl)-2,5-diphenyltetrazolium bromide) and NRU (Neutral Red Uptake)—have the limit of providing results only at defined time points. On the contrary, electric cell-substrate impedance sensing (ECIS), which we have used in the present study, allows to continuously monitor cell viability, thus providing particularly interesting data [14,15]. The technique is based on the monitoring of the electrical impedance presented by electrodes on which cells are seeded and subjected to stimuli of different nature (physical, chemical, biological). Since the impedance changes are only due to the cells, cytotoxic activity can be followed in real time. The ECIS technique has been successfully applied to investigations on several topics, including invasivity of cancer cells, barrier function of endothelial cells and signal transduction involving G protein-coupled receptors (GPCR) for modern drug discovery [16,17].

In this study, the in vitro cytotoxic effect of GA-loaded core-shell NPs (GA-NPs) on HepG2 (human hepatocarcinoma cell line) using ECIS, as well as the correlation between cytotoxicity and GA intracellular concentration were evaluated.

## 2. Results

### 2.1. GA-NPs Characterizations

The morphology of NPs and GA-NPs was investigated by scanning electron microscopy (SEM) and transmission electron microscopy (TEM). A representative image of empty NPs is shown in Figure 1a, where the observed morphology was confirmed to be homogeneous and spheroidal (Figure 1b). GA loading into NPs did not alter significantly their morphology, as can be deduced from Figure 1b and Figure 2a,b.

The hydrodynamic diameters of empty NPs and GA-NPs were measured by DLS (Figure 3). GA loading did not significantly affect NPs diameter. The zeta potential of empty NPs was +19.8 ± 0.1 mV and that of GA-NPs was +19.6 ± 0.1 mV.

GA entrapment efficiency, measured by a spectrophotometric method, was 81%, corresponding to a GA loading of 288 μg/mg of NPs.

### 2.2. In Vitro Release Studies of GA-NPs in Phosphate Buffer Solution

GA release from GA-NPs was investigated within 168 h. In Figure 4 the release profile obtained for GA in phosphate buffer solution (PBS, pH 7.4) is reported. As it can be seen, the release curve has a very steep slope in the first 4 h, then it decreases significantly up to 72 h, at the end of the experiment, when a cumulative GA release of 18–20% is reached.

### 2.3. Determination of Cytotoxic GA Concentration

Dimethyl sulfoxide (DMSO) or empty NPs-treated cells did not show any significant increase in cell mortality at incubation times of 24 h.

Cells incubated with 500 or 350 µmol/L of non vehiculated GA (GA-f) also showed a moderate increase (about 40%) in cell mortality compared to untreated cells. All the other formulations induced a slightly toxic effect (Figure 5). The difference in the toxic effect among the 500 µmol/L GA formulation and all the others was statistically significant, for GA-f as well as for GA-NPs.

Based on the obtained results, all following experiments were carried out using the moderately toxic concentration (500 µmol/L GA) and a slightly toxic concentration (200 µmol/L GA) for both formulations.

### 2.4. Determination of GA-NPs Cytotoxic Effects by Electric Cells-Substrate Impedance Sensing (ECIS)

Figure 6 shows the measurements performed with the ECIS system, obtained by treating HepG2 cells with different GA concentrations, administered both in free form and as GA-NPs. The toxicity of empty NPs was also assessed.

The graph shows on the abscissa axis the elapsed time (in hours) starting from cell seeding of the array and, on the ordinate axis, the normalized resistance provided by the cells-electrode system to the passage of electric current. The time of treatment delivery was chosen as the reference for normalizing each of the curves associated to the experiment.

The increase in resistance values, recorded during the first 24 h, is relative to untreated cells and is due to their adhesion to the substrate (i.e., the electrode) and mitotic divisions. As cells attach to and/or multiply on the well, the passage of current is hindered between the electrode associated to each well and the return electrode. Therefore, in this preliminary phase of the experiments a steady increase of the electrical resistance is expected.

Starting from the twenty-fourth hour, the recordings report the effect of GA 500 and 200 µM (administered in both free and vehiculated form) on cell viability, determined by varying the electrical impedance offered by the cell layers.

The curve relative to the control with unloaded NPs is not markedly different with respect to the control without NPs, only with the medium; this indicates that the unloaded NPs do not present an intrinsic negative effect on the cells. A contrasting result is instead given by the effect of the 200 µM in encapsulated form in comparison with the same dose in free form: the treatment with NPs did not hamper the steady increase of normalized resistance (similarly to the two controls), in continuation of the time course before treatment, whereas the free-form GA treatment was characterized by a flat, or even decreasing, time course of R_norm_, suggesting an interference with the attachment and spreading of cells that would otherwise occur in normal conditions.

As for the treatments with 500 µM, either of the two forms was seen to elicit a negative effect on cell metabolism, with GA-NP giving slightly higher values of normalized resistance than free GA. The result suggests that, even though the encapsulated GA was not capable of reaching the cellular compartments with the same efficacy as the free form of GA (see following Section 2.5, Determination of intracellular concentration of GA), the released GA levels by NPs were high enough to interfere with the cellular functioning in a similar way to free GA. Instead, for the case with 200 µM the remarkable difference between GA-NPs and free GA points out that the dosage obtained with the encapsulated GA is subthreshold for cell toxicity, whereas a toxic effect was clearly manifested by the 200 µM dosage in free form.

### 2.5. Determination of Intracellular Concentration of GA

The intracellular quantity of GA was determined by High Performed Liquid Chromatography (HPLC) after 2 h of incubation with cells. Experiments were carried out in two different conditions, as explained in Section 4.8.

Figure 7a shows the intracellular amount of GA determined in condition 1. In this modality the NPs were not dissolved, therefore part of the GA remained inside them and, in the cytoplasm, only the quantity released in the considered time interval was revealed. As shown in Figure 7a and in Table 1, the intracellular concentration of GA is statistically higher in the non-conveyed form than in the conveyed form.

Panel b shows the results obtained in condition 2. In these conditions, the NPs penetrated inside the cells were lysed (GA-NPs lys) with chloroform releasing all the GA entrapped in them. In this way, it was observed that the NPs were able to transport inside the cells a higher GA amount than what was possible to obtain by diffusion through the cell membrane.

Moreover, it is interesting to note that not all the GA entered, conveyed by NPs, was immediately available for the cells. In fact, after two hours of incubation, only about 5% of the GA present inside the NPs was released into the cytoplasm. In fact, NPs rapidly penetrated inside the cells, but, due to the profile of drug release, the quantity of GA available inside the cells was statically lower than that entrapped within the NPs (Table 1).

## 3. Discussion

The vehiculation of active compounds by using polymeric carriers has numerous advantages: among others, it reduces the drug’s bio-dispersion, modulates its release inside the cells and solves solubility problems [9].

GA is a compound with known antiproliferative properties aimed at cancer cells, and it is also able to interact directly with hepatocytes because it can bind to specific receptors on the surface of these cells, but it is not water-soluble. Due to its lipophilicity, GA needs to be transported in the plasma, but thanks to its capability to recognize hepatocytes it can be used as a target to carry drugs directly inside the liver [6] and, lastly, thanks to its cytotoxic activity, it can be used as an anticancer drug [18]. Moreover, GA has important antibacterial activities, for this reason many studies have evaluated its bactericidal effect conveyed through PLGA NPs [19,20]. In this work, GA was encapsulated in core-shell CS-PLGA NPs and the morphology of the obtained formulations was analyzed by SEM and TEM. The results showed homogenous spherical NPs with a diameter of approximately 280 nm and a zeta potential of ~+20 mV. The release experiments highlighted the possibility to control the amount of bioavailable GA from NPs, succeeding in a prolonged retention profile of the drug. Literature data have also shown a partial release of GA in PBS over time. Darvishi [19] observed 40% release of GA up to 80 h at 37 °C from PLGA NPs obtained with a polymer: drug ratio 1: 1. In our experimental conditions we observed a lower GA release by using CS-coated PLGA NPs with a 2:1 polymer: drug ratio. The results obtained during release experiments can be ascribed both to the different nano formulation of our materials and to the hydrophobicity of GA. In fact, GA solubility in an aqueous buffer, such as PBS, is limited and this is probably the reason for which the amount of released compound was very low.

In any case, we are sure that the low GA release is not an artifact due to its precipitation in PBS, as GA concentration was determined using a calibration curve obtained with standard solutions of GA in PBS with concentrations in the 0.0052—0.04 mg/mL range. This concentration range was chosen because the calibration line had an r^2^ value of 0.997, with a highly linear relationship and higher concentrations were not in line with the absorbance. This last situation, in addition to not allowing a correct evaluation of the concentration, could also be due to a saturation of the GA solution.

Moreover, since the toxic effect of a compound depends on how it is administered (vehiculated by a carrier or in solution), the cytotoxicity caused by GA on a cell line of hepatocellular carcinoma, both free and in encapsulated form, was evaluated.

Since the classic cytotoxicity tests can evaluate the effect of the substances only at fixed times, in this study ECIS was used to evaluate, over time, the effect of GA in both formulations.

Although the effect of GA on HepG2 cells is already known [6], to decide the concentrations of GA to be used in ECIS, an MTT test was performed with different concentrations of GA, both in free and vehiculated form. The obtained results showed a difference in toxicity between the two forms only at a concentration of 350 µmol/L. In this case, GA-f caused a higher toxic effect than GA-NPs.

By using the ECIS technique, the control exhibits the greatest electrical resistance, indicating rapid cell growth in the wells; this effect seems to be increased by the presence of empty nanoparticles. The toxicity induced by GA-NPs 500 µmol/L is similar to that induced by the non-conveyed drug at the same concentration.

Considering the toxicity caused by GA at a concentration of 200 µmol/L, the results obtained during the first 24 h for the conveyed form are similar to those of MTT, while the non-conveyed form shows a decidedly important toxic effect, not observed in MTT. This effect persisted throughout the analysis period.

Moreover, the absence of toxicity of conveyed 200 µmol/L GA formulation is verified by the high similarity of its resistance time course with that of the two controls, for the entire duration of the recordings. The toxicity of 200 µmol/L GA in free form, as given by ECIS measurements, persisted throughout the analysis period. For the first 24 h of the experiment, therefore, a discrepancy between the MTT and ECIS results is observed only as far as the conveyed 200 µmol/L concentration is considered. Apart from the 24 h time point, useful for the comparison with MTT results, the difference between NP and free form at 200 µmol/L was consistently different throughout the ECIS experiment, evidencing a significant difference in the toxic effect of the respective dose delivery.

To try to understand the reasons for these results, the amount of GA available inside the cells in the various formulations was determined. To minimize the effects of hepatic metabolism on the intracellular concentration of GA the measurements were performed after two hours of incubation. The results showed that the amount of GA-NPs inside the cells is much higher than that found when the same amount of GA is administered as GA-f. However, not all GA is released immediately by NPs, but it is instead gradually released, as indicated both by HPLC results and release kinetics studies in PBS. On this basis, GA-NPs 200 µmol/L did not show—in ECIS test- the same toxic effect of GA-f 200 µmol/L because the intracellular GA concentration remains at the sub-cytotoxic level. The critical level for intracellular GA concentration, in encapsulated form, according to both ECIS and HPLC results, can be found within the range 200–500 µmol/L in encapsulated form.

## 4. Materials and Methods

### 4.1. Materials

Poly (D,L-lactide-co-glycolide) (PLGA, lactide:glycolide 50:50, MW 30–60 kDa), chitosan (CS, MW 50–190 kDa), cell culture medium and reagents, chloroform, ethanol (EtOH), tetrahydrofuran (HPLC grade), acetonitrile (CH_3_CN, HPLC grade), methanol (CH_3_OH, HPLC grade) and all other chemicals and solvents were purchased from Sigma-Aldrich (St. Louis, MO, USA), unless otherwise indicated, and used as received. 18-*β*-glycyrrhetinic acid (GA) was purchased from Acros Organics (VWR International Srl, Milan, Italy). Ultrapure water (obtained by a P. Nix Power System apparatus, Human, Seoul, Korea) was used for HPLC analyses.

### 4.2. Synthesis of GA-NPs

GA-loaded PLGA NPs were produced using a one-step osmosis-based methodology [9,19]. 40 mg of PLGA and 20 mg of GA were dissolved in 5 mL of dimethyl sulfoxide (DMSO); the solution was then transferred in a dialysis bag and immersed in 200 mL of water. After 72 h, the precipitated polymer was recovered by centrifugation (14.000 rpm, 20 min), washed three times with water and freeze-dried. GA-loaded PLGA NPs were then coated with CS, as reported previously [21], for the preparation of GA-NPs. A fixed amount of GA-loaded PLGA NPs was suspended in 1% *w*/*v* CS in acetic acid solution at a CS:PLGA ratio of 2:1 (*w*/*w*). The mixture was sonicated for 10 min and then incubated overnight at room temperature, under magnetic stirring. The suspension was then centrifuged at 14.000 rpm for 20 min at 4 °C and the supernatant was removed. The pellet was then washed twice with water, recovered by centrifugation and freeze-dried.

### 4.3. Physico-Chemical Characterization of GA-NPs

GA-NPs morphology was investigated by scanning electron microscopy (SEM, ‘Supra 24’ Zeiss). Dried GA-NPs were mounted onto an aluminum stab using double-sided carbon tape and coated with gold using a sputter coater (Agar Scientific B7234). Moreover, GA-NPs were observed using a transmission electron microscope (TEM, ‘LIBRA 120’ Zeiss), seeding 10 μL of an aqueous dispersion of GA-NPs on carbon-coated copper grids (Formvar Carbon Film 200 mesh copper) according to a previously described method [9].

Dynamic light scattering (DLS) experiments and zeta potential measurements were carried out with a Zetasizer Nano S (Malvern Instruments, Malvern, UK) equipped with a 4 mW He–Ne laser (633 nm). Peak intensity analysis was used to determine the average hydrodynamic diameter of the scattering particles.

The drug content of GA-NPs (loading capacity) was measured using a spectrophotometric method. Precisely measured amounts of GA-NPs (in mg) were dissolved in chloroform and the absorbance of the obtained solution was measured at λ = 247 nm and compared with a calibration curve (Appendix A).

The encapsulation efficiency was calculated according to the following formula:GA entrapment efficiency = [mg GA entrapped/mg GA initial] × 100

### 4.4. In Vitro Release Studies of GA-NPs in PBS

The in vitro release studies were performed by dispersing 3.0 mg of GA-NPs (containing 1.22 mg GA) in (2 mL) of phosphate buffer solutions (PBS, 0.1 M, pH 7.4) under magnetic stirring (300 rpm). At fixed time intervals (1, 2, 4, 24, 48, 72, 144 and 168 h), 1 mL of the supernatant was withdrawn and replaced with fresh PBS [9]. The amount of released GA in the collected samples was determined by measuring their absorbance at λ = 247 nm with a UV–vis Spectrophotometer (Ultraspec 4000, Pharmacia Biotech, Milan Italy). GA concentration in each sample was measured after centrifugation for 5 min at 10,000× *g*, to ensure that no NPs were collected during liquid removal. The obtained absorbance values were compared with a calibration curve, prepared with standard solutions of GA in PBS with concentrations in the 0.0052–0.04 mg/mL range (in Appendix A a calibration curve of GA in PBS utilized to verify the solubility of GA in our experimental conditions is reported).

To verify both the stability of GA during the time and the absence of a strong interaction between GA and NPs an in vitro release study in ethanol (a well-known GA solvent) was performed with the same procedure utilized in PBS. The results are reported in Appendix A.

All experiments were performed in triplicate.

### 4.5. Cell Cultures

HepG2 cells (ECACC, Porton Down, UK) were cultured at 37 °C in a 5% CO_2_ humidified environment in Iscove Dulbecco’s Modified of Eagles’ Medium (IDMEM) supplemented with 10% Fetal Calf Serum (FCS), 500 units/mL penicillin, 10 mg/mL streptomycin, 20 mmol/L L-glutamine, 1% not essential aminoacids (NEAA).

### 4.6. Effect of Different GA Formulations on Cell Viability

DMSO solutions of GA (from 25 mmol/L to 500 mmol/L) were prepared immediately before use. In order to evaluate the GA concentration values able to induce cytotoxicity, HepG2 cells (1 × 10^4^ cells/well) were seeded into a 96-well tissue culture plate (Costar, Cambridge, MA) and cultured for 24 h until a sub-confluent monolayer was formed. Cells were treated with GA administered either free (GA-f) or encapsulated within NPs (GA-NPs); according to the different GA formulations, increasing concentrations of the active drug (in the 25–500 μmol/L range) were obtained by weighing the proper amount of the corresponding formulation and adding it to cell monolayers by changing the culture medium. A final concentration of 0.1% *v/v* of DMSO was utilized in all samples because it did not induce any alterations in cell vitality. To verify the absence of toxic effects due to empty NPs, an amount of these NPs (equal to that used to administer 500 μmol/L GA) was added to the cells. After 24 h of incubation, cell viability was evaluated by the MTT test, according to a previously described protocol [22]. Briefly, 20 μL of a solution of MTT in PBS (phosphate buffer, 5 mg/mL) were added to the medium (0.20 mL) and, after incubation for 4 h at 37 °C, the produced intra-cellular formazan crystals were solubilized with a solution of HCl in isopropanol (4 × 10^−2^ M, 0.20 mL). The optical density (OD) of the solutions in each well was determined using an automatic microplate photometer (ELx800; BioTek, Bad Friedrichshall, Germany) at a wavelength of 570 nm. Each experiment was performed in sextuplicate and repeated four times and the cytotoxicity was calculated according to the following Equation [22]:% cell mortality = [(ODcontrol − ODsample)/ODcontrol] × 100(1)

Specimens were rated as slightly, moderately or severely cytotoxic when the toxic effects, relative to controls, were <30%, between 30% and 60%, or >60%, respectively [23].

### 4.7. Determination of Cytotoxic Effects of GA-NPs by Electric Cells-Substrate Impedance Sensing (ECIS)

The ECIS assay was performed to verify the effect on HepG2 of GA-f and GA-NPs for an extended period. The final GA concentrations utilized were: 200 and 500 µmol/L.

The measurements were carried out at 4000 Hz in order to have the maximum sensitivity of electrical resistance to cell alterations, as indicated by Wegener et al. [24]. 8-well arrays by AppledBioPhysics Inc. (USA) were used in the experiments. In particular, the array model 8WCP PET was used, providing for each well a total electrode area of 3.985 mm^2^ located on inter-digitated fingers to allow measurements of cells. Each well has a substrate area of 0.8 cm^2^ and a maximum volume of 600 μL. On average, with a confluent layer, approximately 4000 to 8000 cells can be measured by the electrodes.

Arrays are oxygen plasma etched by the manufacturer prior to shipment, in order to clean and sterilize the electrodes. During storage of the array, small molecules in the atmosphere can absorb to the electrodes’ gold surfaces, resulting in increased electrode impedance. When exposed to tissue culture medium, after the beginning of the experiment, the desorption of these molecules gradually causes the impedance to return to its original value. In order to avoid this effect, potentially confounding for the interpretation of the electrical changes due to the cells’ activity, the manufacturer recommends to subject the array to electrical stabilization: each well is provided with fresh medium (without cells), and impedance measurements are started, for a sufficiently long time. We chose to carry out this preliminary phase for 24 h; as confirmed by the measurements, this is sufficient for the resistance and capacitance to stabilize at the respective plateau value.

After this step, the cells were seeded in the wells of the array 24 h prior to the treatment [25], and continuously monitored with ECIS in an incubator (37 °C, 95% humidity, 5% CO_2_). The treatments were delivered under biological hood.

On the first day, 500 µL of DMEM were added inside the wells of the ECIS culture array, connected to the ECIS instrumentation. After 24 h, the medium was removed and HepG2 cells (5 × 10^4^ cells/well) were seeded in the wells and cultured for 24 h. Then, cells were treated with the above indicated GA-f and GA-NPs concentrations (or with empty NPs) and monitored for 6 days.

### 4.8. HPLC Determination of Intracellular Concentration of GA

Chromatographic conditions were chosen based on previous work by our group [26]. Briefly: samples were analyzed using a JASCO HPLC system (2 PU-980 pumps, UV-970 UV/VIS detector and AS-1555 autosampler). The analyses were performed at a wavelength of 254 nm with a C-18 (3 μm) Supelco reversed phase column (150 × 4.7 mm using a mobile phase of 80% CH_3_OH (A) and 20% of CH_3_CN: tetrahydrofuran: water (10:80:10, *V/V/V*) (B) (15 min), 1.0 mL/min flow, 50 μL injected volume [26]. The concentration of GA in each sample was quantified using the calibration curve performed with standard solutions before each analysis. Each determination was repeated three times and each experiment was performed 4 times (*n* = 4). HepG2 were plated in 25 cm^2^ flasks in 20 mL of DMEM, at a density of approximately 25,000 cells/cm^2^ and cultured to sub-confluent monolayers; GA-f or GA-NPs were then added to the cells to reach a final GA concentration of 200 μmol/L or 500 μmol/L and incubated for 2 h at 37 °C. Cell monolayers with empty NPs and with DMSO 0,1% were used as controls. After incubation, the cells were washed with PBS solution and lysed by freezing (−80 °C) [26,27]. Cellular lysates were resuspended in 1 mL of H_2_O, centrifuged (20,000× *g*, 15 min, 4 °C) and the supernatants were collected, divided in two aliquots of 500 μL each and evaporated. After evaporation one aliquot was resuspended in 250 μL of H_2_O (condition 1) and the other one in 250 μL of chloroform (condition 2). In the first condition the NPs remain intact and, consequently, the GA determined by HPLC is only the quantity originally released into the cytoplasm by NPs. In the second condition, NPs were lysed by chloroform and therefore their GA content was totally released in the test tube. Thus, in this way, it is possible to determine the total amount of GA inside the cells, even if not yet released by NPs.

### 4.9. Statistical Analysis

All results are expressed as mean ± standard deviation taking into consideration at least three different experiments performed in duplicates. The means were compared by analysis of variance followed by a multiple comparison (if the difference was significant) of means using the Student–Newman–Keuls test. The level of significance was set at 0.05.

## 5. Conclusions

In this work, we used an innovative approach to evaluate the differences in the cellular response to drugs delivered in free and encapsulated form, using a real-time bioimpedance system and comparing the latter’s results with the MTT assay. The data emerging from this study indicate that carrier-based delivery can significantly alter the concentration of the drug within the cells and its release from NPs. Consequently, the delivery systems can modify the toxic effect of the active ingredient, due to the peculiar features of each drug delivery system. Thus, it is very important, particularly when using drug delivery systems, to monitor the toxic effect over time utilizing a real-time test. Our results, moreover, demonstrate the correlation between the toxic effect of the different forms and the bioavailability of the drug.

## Figures and Tables

**Figure 1 materials-14-03893-f001:**
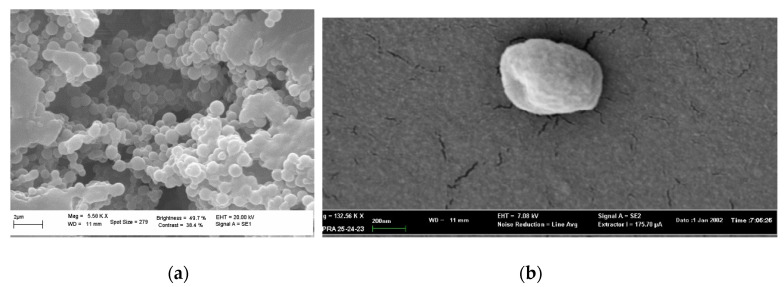
Scanning electron microscopy micrographs. (**a**) Empty NPs are shown (dimension bar: 2 μm); (**b**) a single GA-NP is shown (dimension bar: 200 nm).

**Figure 2 materials-14-03893-f002:**
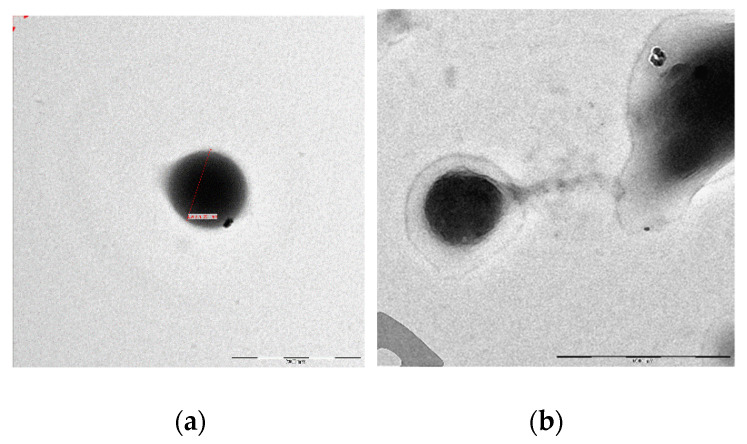
Transmission electron microscopy micrographs. (**a**) a single empty NP is shown (dimension bar: 200 nm); (**b**) a single GA-NP is shown (dimension bar: 500 nm).

**Figure 3 materials-14-03893-f003:**
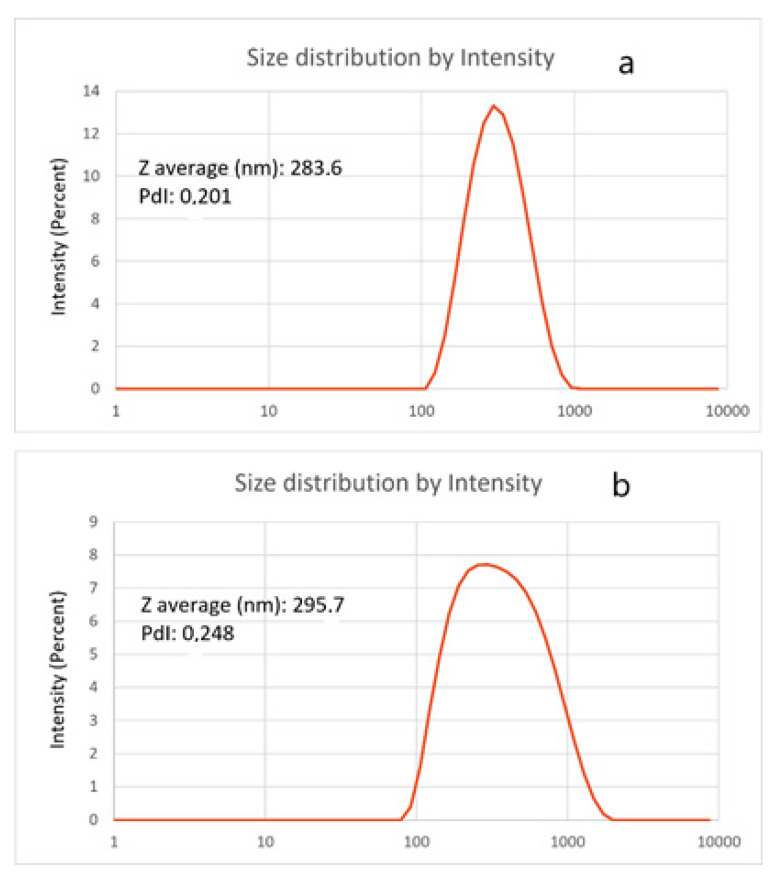
Size distribution by intensity, average diameter and PdI of empty NPs (**a**) and GA-NPs (**b**) obtained by DLS measurements.

**Figure 4 materials-14-03893-f004:**
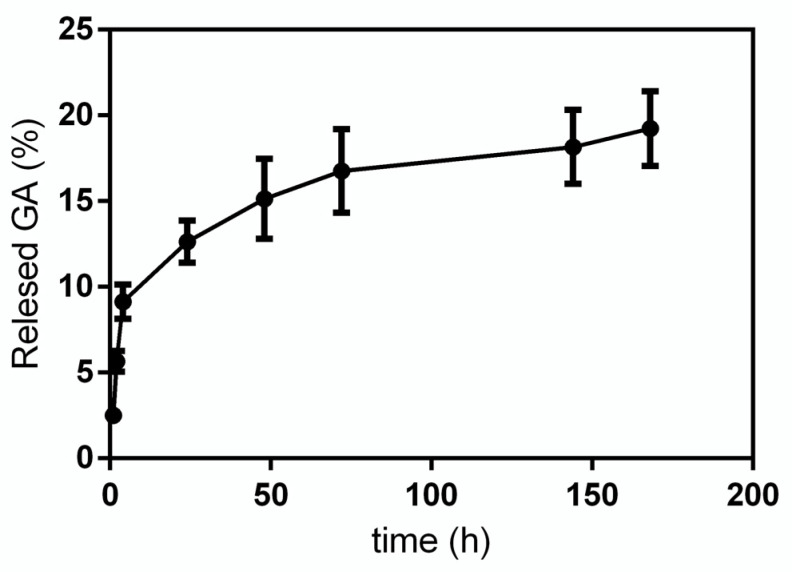
GA release from GA-NPs in PBS.

**Figure 5 materials-14-03893-f005:**
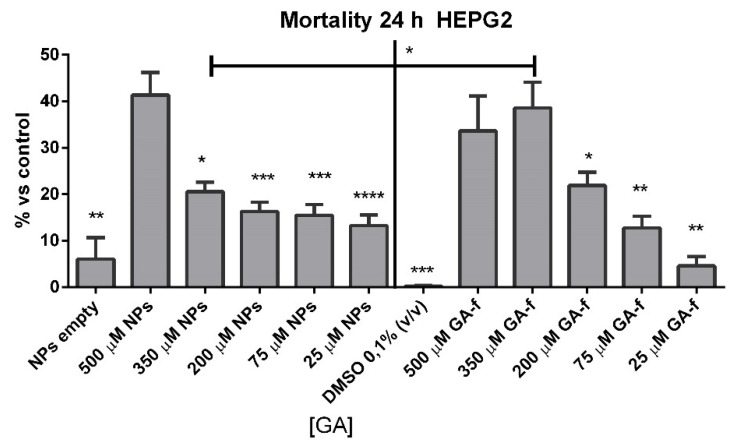
Cytotoxic effects of GA-f and GA-NPs on HepG2 cell line. Each experiment was performed in sextuplicate and repeated three times. **** *p* < 0.0001; *** *p* < 0.001; ** *p* < 0.01; * *p* < 0.05 vs. 500 µmol/L GA.

**Figure 6 materials-14-03893-f006:**
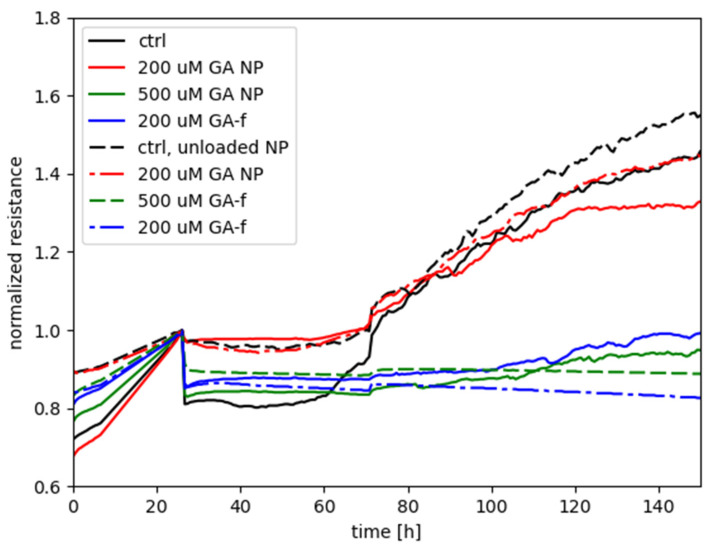
Cytotoxic effects of GA-f and GA-NPs on HepG2 cell line evaluated by ECIS technique.

**Figure 7 materials-14-03893-f007:**
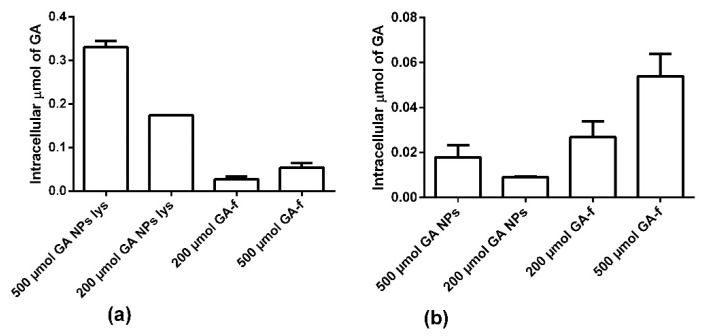
Intracellular concentration of GA (**a**) intracellular amount of GA determined in condition 1. (**b**) the results obtained in condition 2. In these conditions, the NPs penetrated inside the cells are lysed with chloroform releasing all the GA entrapped in them.

**Table 1 materials-14-03893-t001:** Statistical analysis of the results showed in Figure 6.

	*p* Value		*p* Value
500 µmol/L GA-NPs lys Vs. 500 µmol/L GA-NPs	-	200 µmol/L GA-NPs lys	-
	<0.0001	Vs.	<0.0001
	-	200 µmol/L GA-NPs	-
500 µmol/L GA-NPs Vs. 500 µmol/L GA-f	-	200 µmol/L GA-NPs	-
	<0.05	Vs	<0.05
	-	200 µmol/L GA-f	-
	-	-	-

## Data Availability

Data is contained within the article or Appendix A.

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
