# Peer review of "Controlled Release of 18-β-Glycyrrhetinic Acid from Core-Shell Nanoparticles: Effects on Cytotoxicity and Intracellular Concentration in HepG2 Cell Line"

_materials, 2021, doi:10.3390/ma14143893_

Round 1

Reviewer 1 Report

The manuscript submitted by G. Nocca et al. describes a methodology for developing and investigating the in vitro performance of GA-containing NPs in cell-oriented studies. This is an interesting topic, but more details and critical analysis is required in some parts of the manuscript to improve its content. The following comments should be addressed before I would support publication:

General Comment

  1. A major issue is that the authors have not structured their text properly, regarding the provided clarifications for various terms in each section. There are multiple places in the manuscript where a) abbreviations are introduced twice, b) abbreviations have not been introduced, c) used terms are unclear. The reader must constantly go back and forth in order to understand the meaning of each term. Please read the manuscript carefully, and revise where necessary for missing clarifications.

Specific Comments

Introduction

  1. Part of the Introduction section can be revised. There are multiple standalone paragraphs that consist of 1 large sentence, exclusively (Lines 48-86). The paragraphs can be restructured properly, to improve the coherence.

  1. Lines 64-68: References are required for these two sentences.

Results

  1. Line 91: The authors refer to the very low polydispersity of empty NPs. How was it calculated and what is the obtained value?

  1. Lines 89-93: The cohesion of the presented results is disrupted here. The first paragraph mentions the parallel investigations with SEM and TEM. However, the 1st paragraph describes the SEM observations, while TEM observations are merged in the 2nd paragraph with the drug loading studies. Considering that the methodology is described later in the manuscript, the reader assumes that the drug loading has been determined through the TEM studies. Please restructure these lines, so that the related/unrelated studies are mentioned together/separately.

  1. Line 109: This is not a plateau; this is a common behavior for PLGA NPs. Also, please revise the term “kinetics” in the legend of Figure 3 and other places in the manuscript. The release kinetics have not been determined mathematically.

  1. Please carefully read the manuscript, in order to introduce abbreviations at the correct place, e.g., GA-f in line 120 has not been introduced. Also, the term “free” in Figure 4 must be exemplified.

  1. Line 172: What is section 2.8?

Discussion

  1. Lines 195-200 are very vague, and proper citations must be introduced.

  1. Please carefully revise the Discussion section for grammar, syntax, and correct use of main and subordinate clauses. Also, phrases like “and so on” (Line 197), “somewhere between” (Line 253) are subjective and vague, and they must be revised.

Materials and Methods

  1. Please revise the numbering of this section, “Discusion” and “Materials and Methods” sections are both numbered as “3”.

  1. The abbreviations of most of the materials mentioned in Lines 257-264 have already been introduced earlier in the manuscript.

  1. Please provide the methodology for CS coating in Lines 270-272.

  1. Please revise the recurrent statements in the whole manuscript like “have been already reported” and “according to a previously described method”. Instead, you can provide details on the methodology adopted and just cite the related work.

  1. The amounts reported in the in vitro studies are not comparable. All amounts are reported as “mass” until this point of the manuscript. Please add the concentration of GA in PBS standard solution in mass/volume units too, instead of mol/L exclusively (Line 296).

  1. In drug release studies, 50% of the medium was withdrawn for sampling, with a total volume of medium 2 mL. Please discuss this issue and provide a suitable reference. Also, what is the solubility of GA in PBS? Have the authors assured the capacity of the medium’s volume to completely solubilize GA? This is critical to investigate the performance of the formulation in drug release studies and has to be discussed, too.

Author Response

Dear Reviewer,

thanks for your comments. We have edited the manuscript in accordance with your suggestions.

Please see the attached file with a point-by-point replay to your questions.

Reviewer 2 Report

This manuscript evaluated the in vitro feasibility of core-shell nanoparticles based on Chitosan-PLGA for delivering 18-β-Glycyrrhetinic Acid. Overall, the manuscript is well-written, and the methods and results are well organized.  (Comments attached as well)

Here are some comments:

  • Lines 66-68: “Moreover, PLGA and CS have been approved by the US Food and Drug Administration (FDA) and European Medicines Agency (EMA) as effective carriers for drug delivery in humans.” This statement is missing references. It would be helpful to the readers if authors could provide references and additional details about the approval of the materials. Especially, FDA approval details related to Chitosan: any available information related to the chemistry of the approved version of chitosan, product, approved route of administration and the levels approved.
  • Although authors included SEM and TEM images, it would be helpful to add particle size details (Average particle size and size distribution/Polydispersity index-PDI) of the nanoparticles in the manuscript? Providing additional details related to Hydrodynamic radius/size and size distribution measured using DLS would be helpful.
  • Details related to Figure 3. The authors mentioned, “As it can be seen, the release curve has a very steep slope in the first 4 hours, then the inclination decreases significantly up to 72 108 h, at which time the plateau is reached with a cumulative release of 18-20% of GA.” Guidelines generally recommend - The last time point should be the time point where at least 80% of the drug has been released. If the maximum amount released is less than 80%, the last time point should be the time when the plateau of the dissolution profile has been reached. Even after a ~week, only 20% of drug release was observed in the manuscript, experimental justification related to low drug release observed at the last time point is not mentioned in the paper, adding this would be helpful.
    • Show the stability of the drug in the release medium for the studied duration?
    • Is the drug strongly binding to the polymer (PLG or Chitosan) and not getting released?
    • Did the authors study drug recovery to see the mass balance of the drug (released vs unreleased drug) in the in vitro release conditions?
  • Figure 4: Based on the in vitro drug release, low levels of the drug is released from the nanoparticles (12-15%) after 24 hours. What does that mean when the cells are incubated for only 24 hours? How does this release numbers correspond to the concentration numbers of drug – free drug released from the nanoparticles?
  • Figure 6: Same comment as Figure 4, as the incubation time was only 2 h.
  • Does a physical mixture of nanoparticles/drug would show a similar impact as drug-loaded nanoparticles? Provide some literature justification for not including a physical mixture control in the experiments.
  • Section 3.4: “The in vitro release studies were performed by dissolving 3.0 mg of GA-NPs (containing 1.22 mg GA) in (2 mL) phosphate buffer solutions (PBS, 0.1M, pH 7.4) under magnetic stirring (300 rpm). At fixed time intervals (1, 2, 4, 24, 48, 72, 144 and 168 h), 1 ml of the supernatant was withdrawn and replaced with fresh PBS.” How did the authors ensure that the collected medium at each time pull does not include any particles? How the accidental collection of nanoparticles was avoided during in vitro release? Since, if any nanoparticles were collected with a time pull, that would reduce the nanoparticles in the container and could impact the release results of subsequent pulls and overall cumulative release profile.

Author Response

(The authors gave the same response as above.)

Reviewer 3 Report

In this work the authors have synthesized and characterized core-shell NPs made of PolyD-L-lactide-co-glycolide coated with chitosan. The authors propose a simple method for preparing the NPs through an osmosis-based methodology to efficiently entrap 18β-glycyrrhetinic acid. The drug loaded NPs were tested in cellular cultures. I consider that the work is of interest and the study is well organized. However, there is a lack of information regarding the characterization of the particles. There are a few points the authors should address before publication:

Line 43: The authors should avoid expressions like ‘and so on’ .

From line 48 to 63: The sentences in this section of the introduction are fragmented as they contain too many paragraphs. I strongly suggest to rewrite this information without so many paragraphs because is confusing.

Line 89: The morphology of NPs and GA-NPs was investigated by SEM and TEM. If authors present the images, they must present the size of the NPs measured through them. For example using the software ImageJ.

Line 102: The scale bar in figure 2 is not very visible, must be improved.

Line 265: Regarding the synthesis of CS-PLGA NPs, section 3.2 - The protocols presented for the preparation of the coating with CS appear to be quite minimal. The type of equipment used for the coating is not indicated.

Line 273: Although the authors said that the thermal and mechanical properties of GA-NPs have been already reported in another study, they must provide more information regarding the characterization of the NPs. They only present SEM and TEM results. How about zeta potential measurements/surface charge of the NPs with and without drug? And specific surface area of the NPs?

Line 281: The authors said “The drug content of GA-NPs (loading capacity) was measured using a spectrophotometric method [9]. Precisely measured amounts of GA-NPs (in mg) were dissolved in chloroform and the absorbance of the obtained solution was measured at λ = 247 nm and compared with a calibration curve.” They should provide the calibration curve and also the UV-VIS spectra (in supporting information).

Line 385: The conclusions are very poor and do not support the results.

Author Response

(The authors gave the same response as above.)

Round 2

Reviewer 1 Report

The authors have successfully addressed this reviewer's comments.

Reviewer 2 Report

NA

Reviewer 3 Report

The paper should be published due to its originality and due to the promising results of potential application.